# Variational Model Perturbation for Source-Free Domain Adaptation

**Mengmeng Jing**[1,2][*] **Xiantong Zhen**[2] [†]**, Jingjing Li**[1]**, Cees G. M. Snoek**[2]
[1]University of Electronic Science and Technology of China
[2]University of Amsterdam

## Abstract

We aim for source-free domain adaptation, where the task is to deploy a model pre-trained on source domains to target domains. The challenges stem from the distribution shift from the source to the target domain, coupled with the unavailability of any source data and labeled target data for optimization. Rather than fine-tuning the model by updating the parameters, we propose to perturb the source model to achieve adaptation to target domains. We introduce perturbations into the model parameters by variational Bayesian inference in a probabilistic framework. By doing so, we can effectively adapt the model to the target domain while largely preserving the discriminative ability. Importantly, we demonstrate the theoretical connection to learning Bayesian neural networks, which proves the generalizability of the perturbed model to target domains. To enable more efficient optimization, we further employ a parameter sharing strategy, which substantially reduces the learnable parameters compared to a fully Bayesian neural network. Our model perturbation provides a new probabilistic way for domain adaptation which enables efficient adaptation to target domains while maximally preserving knowledge in source models. Experiments on several source-free benchmarks under three different evaluation settings verify the effectiveness of the proposed variational model perturbation for source-free domain adaptation.

## 1 Introduction

A well-trained neural network can achieve satisfactory performance on in-distribution tasks. However, in real applications, a common situation is that the training set and the test set are from different distributions. In this case, the performance of the models usually degrades significantly due to domain shifts. Domain adaptation (DA) [1, 2, 3, 4] is proposed to solve this problem by transferring knowledge from the labeled source domain (the training set) to the unlabeled target domain (the test set). It is assumed that source data are still available when adapting to target data. In this paper, we focus on the more challenging scenario where source data are not accessible, which is known as source-free domain adaptation (SFDA) [5, 6]. SFDA promises much wider ranges of applications than the vanilla domain adaptation in that it can circumvent the problem of not disclosing source data in order to protect the privacy of individuals or the intellectual property of commercial entities.

To deal with domain shift, two popular techniques in the literature to adapt the SFDA model to the target domain are fine-tuning (updating the whole parameters) and modulating (tuning the batch normalization layers and freeze all the other parameters). The former focuses on the difference of domains, while the latter focuses on the relevance of domains. For fine-tuning methods, e.g., SHOT [7] and NRC [8], the parameters of the entire model are further updated by designing

---

[*]This work was done when Mengmeng Jing was a visiting student at University of Amsterdam.
[†]Currently with United Imaging Healthcare, Co., Ltd., China.

36th Conference on Neural Information Processing Systems (NeurIPS 2022).

optimization objectives. For example, SHOT [7] calibrates the model predictions by information maximization and pseudo-labeled self-supervision, while NRC [8] exploits the nearest neighbor relationship between samples. Though being empirically successful, both model fine-tuning and batch normalization modulation suffer from drawbacks. Fine-tuning may cause some problems. First, the model will deviate from the source data, leading to performance degradation on the source domain, similar to catastrophic forgetting in continual learning [9, 10, 11, 12]. We care about the performance on the source domain since in the more generalized SFDA scenario [13] we do not necessarily have the target domain ID available, and therefore we would not know when to deploy the source model. Second, fine-tuning could result in a model being biased to a specific target domain (overfitting), thus reducing the generalization of the model [14, 15, 16]. Third, since there are no regularizations from the labeled source data, e.g., cross entropy loss, errors may accumulate and the parameters of the model may be distorted, leading to low performance in Continual Online SFDA [17]. Batch normalization modulation approaches [18, 19, 20] keep weights of all convolutional and fully-connected (FC) layers frozen, and only modulate the statistics in the batch normalization layers [21] to adapt to target data. However, they tend to be less flexible and would not continuously adapt to target data with large domain shifts. In addition, those are all deterministic approaches which suffer from lack of generalizability [22, 23, 24].

In this paper, we propose variational model perturbations, which fixes the existing weights and slightly perturbs them to encode the uncertainties according to the domain shifts. We learn to perturb model parameters to achieve domain adaptation by variational inference. By variational perturbations, we introduce uncertainties into the model. As a result, we perturb a deterministic network into a Bayesian neural network (BNN), allowing the model to generalize to relevant domains. We demonstrate that by perturbing the model in a variational manner, we are equivalent to learning a special BNN with a fixed mean and a learnable variance. To achieve more efficient perturbation, we further adopt a parameter sharing strategy, which significantly reduces the learnable parameters. This enables more efficient adaptation in contrast to learning a full BNN. It is worth noting that the perturbation is naturally regularized due to the variational inference formulation, which prevents the perturbed weights from being distorted during the continuous unsupervised learning. Our approach is easy to implement, offers a plug-and-play module to adapt model parameters and works seamlessly with different optimization objectives. We conduct extensive experiments on five datasets with three different evaluation settings. The results consistently demonstrate the effectiveness of variational model perturbation for source-free domain adaptation.

## 2   Methodology

We start from the vanilla unsupervised domain adaptation, based on which we formally introduce the setting for source-free domain adaptation. For the vanilla unsupervised domain adaptation setting, we are given a source domain $\mathcal{D}_s = \{(x_i^s, y_i^s)|x_i^s \in \mathcal{X}_s, y_i^s \in \mathcal{Y}_s\}_{i=1}^{n_s}$ with $n_s$ labeled data and a target domain $\mathcal{D}_t = \{x_j^t|x_j^t \in \mathcal{X}_t\}_{j=1}^{n_t}$ with $n_t$ unlabelled data under the condition that $p(\mathcal{X}_s) \neq p(\mathcal{X}_t)$ where $p(\mathcal{X}_s)$ and $p(\mathcal{X}_t)$ are the source and target data distributions. The source and target domains share the same label space, i.e., $\mathcal{Y}_s = \mathcal{Y}_t$. The task is to learn a model $f : \mathcal{X}_t \to \mathcal{Y}_t$ that could predict the target labels. In the source-free domain adaptation setting, we are only given the unlabeled target domain data $\mathcal{D}_t = \{x_j^t\}_{j=1}^{n_t}$ and the model trained on the source domain parameterized by $w_s$.

**Model Perturbation with Uncertainty**   From the probabilistic perspective, we define a model $p(w_s|D_s)$ to be trained on the source domain $D_s$. Then, the model parameters $w_s$ can be optimized by maximum likelihood estimation under the i.i.d. assumption:

$$w_s = \arg \max_{w_s} \log p(D_s|w_s) = \arg \max_{w_s} \sum_i \log p(y_i^s|x_i^s, w_s). \tag{1}$$

Usually we could optimize Eq. (1) through mini-batch based stochastic gradient descent [25] and the optimum of $w_s$ is a point estimate of the model parameter. Having a model pre-trained in Eq. (1) on a source domain, we would like to deploy it on different target domains. However, usually the performance of the source model degrades significantly on the target domains due to the domain shift.

Therefore, we propose to perturb the model parameters based on the point estimate $w_s$ in order to make it generalizable to the target domain while maximally maintain the discriminative ability of the source model. Denote the perturbed weights as $w_t$, which is obtained by a simple perturbation

operation as follows:

$$w_t = w_s + \Delta w, \tag{2}$$

where $\Delta w$ is the perturbation for $w_s$. It is worth noting that we only perturb the weights of the convolutional and FC layers in the network and we update the running mean and running variance in the batch normalization layers through moving average.

In this paper, we formulate the perturbation as a variational inference problem in a probabilistic framework. Given the target domain $D_t$, we compute the posterior distribution of the perturbations by Bayesian inference, i.e., $p(\Delta w | D_t)$. However, directly computing the expectations of the posterior distribution is intractable, since it is equivalent to computing expectations of an uncountably infinite set of neural networks [23, 26]. Therefore, as a more realistic alternative, we resort to computing the posterior distributions through variational inference.

Specifically, assume the perturbations are drawn from a zero-mean distribution, i.e., $\Delta w \sim \mathcal{N}(0, \sigma^2 I)$, where $\sigma$ is the learnable standard deviation. Then, our task reduces to solving the optimization problem such that the variational posterior $q(\Delta w | 0, \sigma)$ can approach the true posterior $p(\Delta w | D_t)$. We minimize the Kullback-Leibler (KL) divergence between them as follows:

$$
\begin{aligned}
\sigma^* &= \arg\min_{\sigma} \mathrm{KL}[q(\Delta w|0,\sigma)||p(\Delta w|D_t)] \\
&= \arg\min_{\sigma} \int q(\Delta w|0,\sigma)\log\frac{q(\Delta w|0,\sigma)}{p(\Delta w|D_t)}d\Delta w \\
&= \arg\min_{\sigma} \int q(\Delta w|0,\sigma)\log\frac{q(\Delta w|0,\sigma)}{p(D_t|\Delta w)p(\Delta w)}d\Delta w \\
&= \arg\min_{\sigma} \mathrm{KL}[q(\Delta w|0,\sigma)||p(\Delta w)] - \mathbb{E}_{q(\Delta w|0,\sigma)}[p(D_t|\Delta w)],
\end{aligned}
\tag{3}
$$

where the first term in Eq. (3) is the KL divergence between the posterior and the prior, and the second term is the expectation likelihood. The objective function of our method is Eq. (3). For the expectation likelihood term, it has different implementations and we will discuss it later. The specific forms of the objective function under different implementations are included in supplementary material.

Choosing zero as the mean of $\Delta w$ is because we want to avoid the perturbed model deviating much from the source model so that it can adapt to more general settings, i.e. Generalized SFDA [13] and Continual Online SFDA [17]. In this way, the weights of the perturbed model will vary around the weights of the source model while being generalizable due to the uncertainty induced by non-zero variance. In essence, we expand the effective coverage of the source model to the target domain.

**Connection to Learning a Bayesian Neural Network**    It has been proven that introducing uncertainty into weights can improve generalizability of neural networks [23, 24]. Actually, under mild conditions, our method is equivalently learning a BNN. We demonstrate this as follows.

Assume that we would like to learn a neural network by variational Bayesian inference, where we place a prior $p(w_t)$ over the weight. We design the variational posterior as $q(w_t|w_s, \sigma)$. Here we assume the mean $w_s$ to be a constant which is the point estimate of the model parameters, and the variance $\sigma^2$ to be a learnable variable. We minimize the KL divergence between the variational posterior $q(w_t|w_s, \sigma)$ and the true posterior $p(w_t|D_t)$:

$$\sigma^* = \arg\min_{\sigma} \mathrm{KL}[q(w_t|w_s,\sigma)||p(w_t|D_t)] \tag{4}$$

$$= \arg\min_{\sigma} \int q(w_t|w_s,\sigma)\log\frac{q(w_t|w_s,\sigma)}{p(w_t)p(D_t|w_t)}dw_t. \tag{5}$$

As we define $w_t = \Delta w + w_s$, the posterior $w_t \sim q(w_t|w_s, \sigma^2)$ is transformed from $\Delta w \sim q(\Delta w|0, \sigma^2)$. By adopting the technique of change of variables to Eq. (5), we arrive at:

$$\sigma^* = \arg\min_{\sigma} \int q(\Delta w|0,\sigma)\log\frac{q(\Delta w|0,\sigma)}{p(\Delta w)p(D_t|\Delta w)}d\Delta w \tag{6}$$

$$= \arg\min_{\sigma} \mathrm{KL}[q(\Delta w|0,\sigma)||p(\Delta w|D_t)]. \tag{7}$$

Therefore, minimizing the KL divergence between $q(w_t|w_s, \sigma)$ and $p(w_t|D_t)$ is equivalent to minimizing the KL divergence between $q(\Delta w|0, \sigma)$ and $p(\Delta w|D_t)$.

Notably, in BNN, the prior is usually set to be noninformative normal Gaussian distribution [22, 23, 27]. Here we just assume we adopt the same noninformative normal Gaussian prior, i.e., $p(w_t)=p(\Delta w)$. For practical optimization, since the prior only serves as a regularizer in the KL divergence term while our goal is to infer the posteriors, the two inferred posteriors do not necessarily have the same variance. Even if the prior distributions are different, the conclusion that variational model perturbation is equivalent to learning a BNN still holds.

**Variational Posterior of Perturbations**    Following [23], we assume that the variational posterior distribution is a diagonal Gaussian distribution. Then we can obtain a sample of the perturbations $\Delta w$ by sampling the unit Gaussian and scaling it by the standard deviation $\sigma$. To ensure that $\sigma$ is non-negative, in practical implementations $\sigma$ is not calculated and updated directly. Specifically, $\sigma$ is calculated as $\sigma=\sqrt{\exp(\rho)}$, where $\rho$ is the *de facto* variational posterior parameter which is the logarithm of the variance. Thus, a sample from the perturbed $w_t$ can be represented as:

$$w_t = w_s + \epsilon \circ \sqrt{\exp(\rho)}, \tag{8}$$

where $\epsilon \sim N(0, I)$ and $\circ$ denotes element-wise multiplication.

**Prior of Perturbations**    Previous works have shown that the selection of different priors affects the generalization performance of the model [28, 22, 29]. Considering that weights of each layer have different variations, it may be sub-optimal to use a uniform prior for the whole network. Therefore, we propose to specify different priors for perturbations of each layer according to the statistical properties within the pre-trained model. Specifically, each convolutional layer contains multiple convolutional kernels. Previous research [28] has found that weights in the same convolutional kernel have strong correlations, while different convolutional kernels do not, even if they are in the same layer. In view of this, we propose an *adaptive prior*, which computes the standard deviation of each convolutional kernel in the pre-trained model as the prior of the perturbations of the whole convolutional kernel. Then, the prior shared by the whole convolutional kernel is as follows:

$$p(\Delta w^{li}) = \mathcal{N}(0, \lambda \text{Var}(w_s^{li})I), \tag{9}$$

where $p(\Delta w^{li})$ represents the prior distribution for perturbations of the $i$-th convolution kernel in $l$-th convolutional layer, and Var is the variance of the convolutional kernel in the pre-trained model $w_s$. $\lambda$ is a scale coefficient. We experimentally observed that $\lambda=1.0$ can achieve satisfactory performance.

**Likelihood Function**    In general, variational model perturbation is independent of the likelihood function. In SFDA, the target data are unlabeled, the likelihood function can be implemented by optimization with respect to unsupervised losses. For example, we can minimize entropy loss [18] to encourage the model predictions to be close to one-hot encodings. In addition, Liang et al. [7] highlight that ideal predictions should have enough category diversity, we can achieve this by maximizing the mean of the entropy of predictions in a mini-batch. All these objective functions can be well integrated within our framework. We report the results of our method under different implementations of the likelihood in the experimental section.

**Parameter Sharing Strategy**    Since the parameters in the same convolutional kernel have strong correlations [28], it is a natural choice to adopt a parameter sharing strategy for the convolutional layers so that we can further reduce the learnable parameters, i.e., all the weights in the convolutional kernel share the same perturbations. Thus, the number of learnable parameters for each convolutional layer is equal to the number of output channels. By doing so, we can largely reduce learnable parameters without significantly degrading performance.

Through the parameter sharing strategy, we deconstruct the training process of SFDA into a static *invariant part* and a dynamic *varying part*: we can train a model with millions of parameters (*static part*) in the source domain, and then adapt it to different target domains by learning a small number of perturbation parameters (*varying part*), which avoids training the parameters of the whole model every time, reduces the storage space and makes the process of domain adaptation more flexible.

## 3  Related Works

**Unsupervised Domain Adaptation.**    Unsupervised domain adaptation [30, 31, 32, 33] aims at transferring knowledge from the source domain to the unlabeled target domain in order to perform

Table 1: Difference between our model perturbation and related methods. Model perturbation is the only source-free online domain adaptation method that introduce uncertainty to the model without any extra requirement.

| | Source-Free | Online | Uncertainty | Adaptation Scheme | Extra Requirement |
|---|---|---|---|---|---|
| Tent [18] | √ | √ | × | - | × |
| SHOT [7] | √ | × | × | Conv + FC | × |
| TTT [42] | × | √ | × | Conv + FC | auxiliary task |
| BACS [43] | √ | √ | √ | Conv + FC | multiple models |
| **This paper** | √ | √ | √ | Perturbations | × |

as well on the target domain. Most methods minimize the expected error of the model on the target domain by reducing the domain shifts. For example, methods based on adversarial generative networks [34, 1, 35] learn domain-indistinguishable features in an adversarial manner. Statistical matching methods [36, 37, 38, 39] try to mitigate domain shifts by aligning the first- or second-order statistics between domains.

**Source-Free Domain Adaptation.**   Unsupervised domain adaption assumes that source data is available when adapting to target domains, which would not always hold in real-world applications due to issues of intellectual property, individual privacy, decentralisation, etc. Therefore, Source-Free Domain Adaptation (SFDA) is proposed to solve this problem [5, 6, 7, 8, 40, 41]. In the setting of SFDA, only the pre-trained model in the source domain and the unlabeled target data are provided. Kundu et al. [6] align the source and target domains using generative models. Ishii and Sugiyama [41] reduce domain shifts by aligning the statistics stored in the batch normalization layers. From an information-theoretic perspective, Liang et al. [7] proposes SHOT to fine-tune the source model using target data by imposing entropy minimisation and diversity maximisation for predictions. In addition, SHOT employs self-supervised learning using pseudo-labels. Yang et al. [8] exploit the intrinsic neighborhood structure to encourage label consistency among target data. Huang et al. [40] maintain the consistency of features between the historical model and adapted model through contrastive learning.

**Online Source-Free Domain Adaptation.**   SFDA methods [7, 6, 8, 40] assume all target domain data are available during training, and they also optimize dedicated objective functions through multiple iterations. To be applicable to scenarios such as autonomous driving and real time object tracking online SFDA, also known as test-time adaptation, has been explored. For example, AdaBN [19] and Test-time BN [20] improve the performance of the model in the target domain by recalculating the Batch Normalization (BN) statistics in the test period instead of using fixed BN statistics from the pre-trained model. Tent [18] modulates the parameters in the BN layers through entropy minimization to adapt the test data in real time. For stable and effective adaptation, Tent freezes all the parameters except the BN layers. Test-Time-Training [42] introduces additional auxiliary tasks to improve test-time performance. Cotta [17] adopts weight-averaging and augmentation-averaging to calibrate pseudo-labels to avoid error accumulation during continual test-time adaptation, then they use these reliable pseudo-labels for self-supervised regularization. By contrast, our model perturbation freezes all weights except the batch normalization parameters in the source model, which avoids error accumulation and knowledge forgetting.

We provide a comparison between the proposed model perturbation and related methods in Table 1. Most existing approaches are developed in a deterministic manner, without exploring the uncertainty. Differently, in this work we address source-free domain adaption in a probabilistic framework by modelling weight uncertainty.

**Bayesian Neural Networks.**   Our work is also closely related to Bayesian neural networks. They combine neural networks with Bayesian inference, which explain the uncertainties from the model and provides the distributions over the weights and outputs. Various techniques have been explored to learn Bayesian neural networks including variational inference [44, 26], probabilistic backpropagation [45], Hamiltonian Monte Carlo [46] and the Laplace approximation [27]. Our variational model perturbation provides an alternative way of implementing Bayesian inference for neural networks. Compared to previous implementations, our method is plug-and-play and more flexible. It provides

an efficient way to adapt a deterministic source model to a target domain by transforming into a probabilistic one while avoiding re-training a full Bayesian neural network.

## 4 Experiments

### 4.1 Datasets and Settings

**Datasets.** We report our experiments on five datasets. The **Office** [47] dataset includes 3 domains. All the images are categorized into 31 classes. **Office-Home** [48] consists of 4 domains. Each domain contains 65 different classes. **CIFAR10-C**, **CIFAR100-C** [49] and **ImageNet-C** [49] include 15 different types of corruptions imposed on the original CIFAR10, CIFAR100 and ImageNet datasets, respectively. Each corruption type contains 5 different corruption severities. Each severity has 10,000 images in CIFAR10-C/CIFAR100-C and 50,000 images in ImageNet-C, respectively.

**Evaluation Settings.** We test our method on three source-free domain adaptation settings. For a clearer elaboration of the experimental details, we clarify SFDA consists of three phases, namely, source pre-training, target adaptation and testing. *Offline SFDA* [6, 5]: In the target adaptation phase, we can access data of the entire target domain and optimize the objective functions with multiple iterations. *Generalized SFDA* [13]: we split the source data into 80% and 20% parts. In the source pre-training phase, we use the labeled 80% part to pre-train the source model. In the target adaptation phase, we use all the unlabeled target data to adapt the model. In the testing phase, we predict the remaining 20% source data and all target data. We compute the harmonic mean of the source and target accuracies: $\text{Harmonic} = \frac{2*\text{Acc}_S*\text{Acc}_T}{\text{Acc}_S+\text{Acc}_T}$. *Continual Online SFDA* [17]: the target domain phase and the testing phase are carried out simultaneously. At each iteration, we can only access data in one batch and data in the previous iteration cannot be accessed again. We carry out this experiment in a more challenging setting where data of different domains are encountered continually.

The detailed training processes of our method under the three settings are in suplementary material.

### 4.2 Implementation Details

The objective function in Eq. 3 includes the KL divergence loss and the expectation likelihood loss. In the three settings, we adopt different implementations of the expectation likelihood loss.

For Offline SFDA and Generalized SFDA, we employ the information loss and self-supervised loss as in SHOT [7] to optimize the expected likelihood loss. For a fair comparison, we use ResNet-50 as the backbone just as the previous methods [7, 1, 13, 40]. In the source-pretraining phase, this backbone is trained on the source domain to obtain the source pre-trained model. As for the optimizer, following [7], we employ Stochastic Gradient Descent with weight decay 1e-3 and momentum 0.9. As for the learning rate, we set 2e-3 for the backbone model and 2e-2 for the bottleneck layer newly added in SHOT. In addition, $\beta$ is a hyper-parameter which balances the importance of information loss and self-supervision loss in SHOT. Since the parameters learned by model perturbations are different from those of SHOT, the value of $\beta$ used in SHOT may not be optimal. Therefore, we search the optimal value $\beta^*$ in range of [0.1,2.0] for all the three datasets. Then, $\beta^*$ is set to 0.3 in both Office and Office-Home. The batch size is 64.

For Continual Online SFDA, we use the entropy loss as the likelihood function in Eq. 3. In this way, the implementation is equivalent to a perturbation of Tent [18]. We use the same network architectures as in [17] to make the results comparable. In CIFAR10-C, we employ WideResNet-28 [50] from Robust Bench [51] as the backbone network. We used the Adam [52] optimizer with learning rate 5e-4. In CIFAR100-C tasks, we used ResNeXt-29 [53] from [54]. The optimizer settings are the same as that in CIFAR10-C except the learning rate is 7e-5. In the ImageNet-C experiments, a pre-trained ResNet-50 is employed. We adopt the Stochastic Gradient Descent optimizer with learning rate 1e-4, momentum 0.9 and weight decay 0. In the testing phase, given a batch of target samples, we iterate the proposed method for only one epoch and make predictions immediately.

We set the scale parameter $\lambda$ of the prior to be $1.0$. We discuss the impact of different choices of priors on performance in Section 4.3. In the target adaptation phase, we sample the perturbations once for running efficiency, while in the test phase, we sample the perturbations 10 times. To reduce the sampling variance, we employ the local reparameterization trick [55]. The detailed optimization

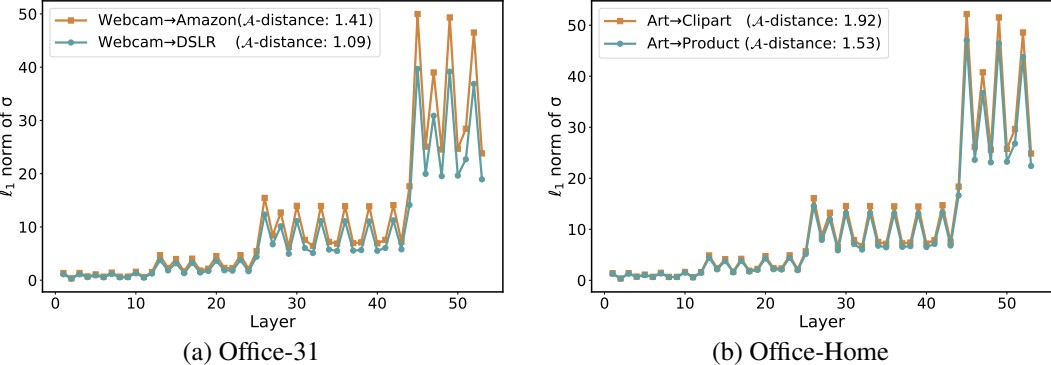

(a) Office-31                        (b) Office-Home

Figure 1: We measure the weight uncertainty by an $\ell_1$ norm of $\sigma$ per ResNet-50 layer. The $\sigma$ for a difficult task (e.g., W→A) has a larger $\ell_1$ norm than the easy task (e.g., W→D), and correspondingly the perturbations vary over a larger range, i.e., a larger uncertainty. Therefore, the perturbations we learned can be regarded as a measure of uncertainty, which reflects the distribution shifts between domains. Here we use the $\mathcal{A}$-distance as an indicator of the domain gaps, where a larger $\mathcal{A}$-distance means a larger domain gap.

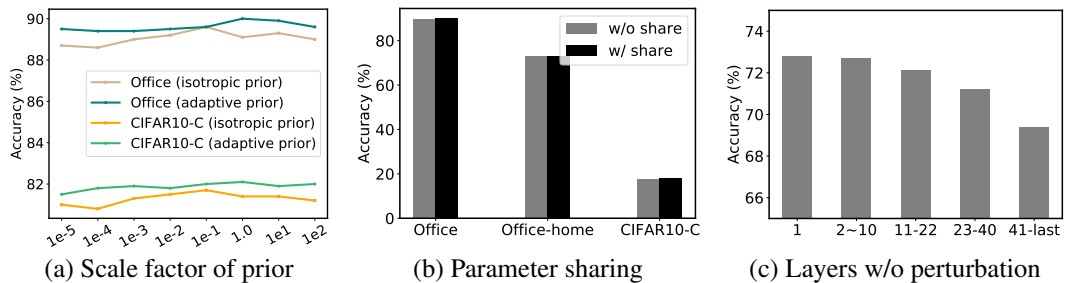

(a) Scale factor of prior       (b) Parameter sharing       (c) Layers w/o perturbation

Figure 2: (a) Impact of different variance scale factors; (b) Comparison of performance with and without parameter sharing strategy. "w/o share" means we learn a different $\sigma$ for each parameter in the convolutional kernel individually; (c) Impact of layers without perturbations where the results are based on Office-Home dataset and the backbone is ResNet-50.

process with local reparameterization trick is in the supplementary material. We also discuss the effect of different numbers of Monte Carlo samples on performance in the training phase in the supplementary material.

As for the compared methods, we either report the results in the original papers or the best results we can reproduce. For the non-Bayesian methods, we run them 10 times with different random seeds and take the average of them to reduce randomness. Apart from SHOT and Tent, we have also applied our variational model perturbation to other methods and report the results in the supplementary material.

Code is available at `https://github.com/mmjing/Variational_Model_Perturbation`.

### 4.3 Effectiveness Analysis

We present an effectiveness analysis of our method in Fig. 1, Fig. 2 and Fig. 3. Results of our method in the offline setting are from perturbed SHOT and in the online setting from peturbed Tent.

**Quantification of Weight Uncertainty.** In Fig. 1 we report a proxy of the weight uncertainty using the $\ell_1$-norm of the learned $\sigma$ per layer. We make the following observations: (1) In the first few layers of the network, the perturbations are small. As the layers go deeper, the perturbations become larger. This shows that the shallower layers of the neural networks are more generalized, while the deep layers are task-specific, which is consistent with the findings in previous work [56, 38]; (2) The extent of the perturbation is task-dependent. A larger domain shift of the task needs a larger perturbation, which demonstrates that model perturbation effectively encodes uncertainty into the network, thus enhancing the generalization.

Table 2: Offline SFDA accuracy (%) on Office, where all methods use a ResNet-50 backbone. Our variational model perturbation performs best on the majority of target domains and results in the best average accuracy.

| | Office | | | | | | |
|---|---|---|---|---|---|---|---|
| | A→W | A→D | W→A | W→D | D→A | D→W | Mean |
| **Source-use** | | | | | | | |
| DAN [38] | 80.5 | 78.6 | 62.8 | 99.6 | 63.6 | 97.1 | 80.4 |
| DANN [57] | 82.6 | 81.5 | 67.5 | 99.3 | 68.4 | 96.9 | 82.7 |
| MCD [34] | 88.6 | 92.2 | 69.7 | **100.0** | 69.5 | 98.5 | 86.5 |
| BNM [58] | 91.5 | 90.3 | 71.6 | **100.0** | 70.9 | 98.5 | 87.1 |
| SAFN [59] | 90.1 | 90.7 | 70.2 | 99.8 | 73.0 | 98.6 | 87.1 |
| CDAN+E [1] | **94.1** | 92.9 | 69.3 | **100.0** | 71.0 | 98.6 | 87.7 |
| MDD [60] | 90.4 | 90.4 | 73.7 | 99.9 | 75.0 | 98.7 | 88.0 |
| BDG [61] | 93.6 | 93.6 | 72.0 | **100.0** | 73.2 | **99.0** | 88.5 |
| **Source-free** | | | | | | | |
| Source-only | 76.9 | 80.8 | 63.6 | 98.7 | 60.3 | 95.3 | 79.3 |
| SHOT [7] | 90.1 | 94.0 | 74.3 | 99.9 | 74.7 | 98.4 | 88.6 |
| **This paper** | 93.3 | **96.2** | **76.9** | **100.0** | **75.4** | 98.6 | **90.0** |

**Adaptive Prior.** In Fig. 2 (a) we report the impact of the variance scale factor of the prior on performance. Compared to the isotropic Gaussian prior, our adaptive prior has better performance and smaller fluctuations for different scale factors. This indicates that it is reasonable to use an adaptive prior as different weights may have different variation ranges when adaptation. Our adaptive prior can specify different prior distributions for different weights according to the statistical properties within the convolutional kernels, leading to a better performance.

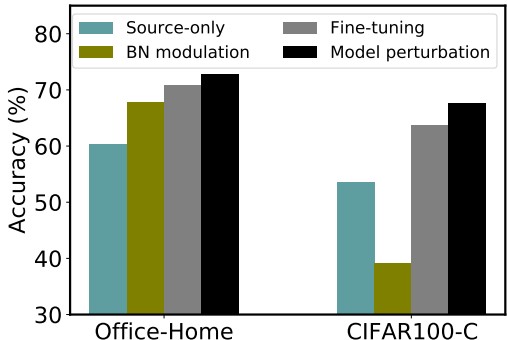

Figure 3: Comparison of different adaptation schemes. Model perturbation outperforms BN modulation and fine-tuning on both offline tasks (Office-Home) and online tasks (CIFAR100-C).

**Parameter Sharing Strategy.** In Fig. 2 (b) we observe that the parameter sharing strategy achieves almost exactly the same performance compared with the non-sharing strategy. For ResNet-50 based tasks, e.g., Office and Office-Home, the learnable parameters reduce from 24.03 million to 600 thousand compared with the non-sharing strategy. As for the WideResNet-28 based tasks, e.g., CIFAR10-C, the learnable parameters reduce from 36.47 million to 30 thousand. These results verify the effectiveness of the parameter sharing strategy.

**Which Layers are Crucial for Perturbation?** We test the performance by keeping some of the network layers unperturbed. As shown in Fig. 2 (c), we observe that the performance of the model degrades considerably when removing perturbations in the last few layers of the network, revealing that perturbations for the last few layers are crucial, since these layers are often task-relevant.

**Adaptation Schemes.** In Fig. 3, we compare results of different updating schemes on Office-Home and CIFAR100-C. BN modulation means only updating the statistics in the batch normalization layers. Fine-tuning represents updating all the weights in the model. We observe that model perturbations outperforms BN modulation and fine-tuning in both the offline and online SFDA tasks. Compared with BN modulation, model perturbation is more flexible. It can generalize the model to more target domains through perturbations. Compared with Fine-tuning, model perturbation avoids the distortion of weight in the source model, which prevents the knowledge in the source domain from being forgotten in continuous adaptation. Therefore, model perturbation is the better choice in SFDA tasks.

Table 3: Offline SFDA accuracy (%) on Office-Home, where all methods use a ResNet-50 backbone. Our variational model perturbation performs best on the majority of target domains and results in the best average accuracy.

| | Office-Home | | | | | | | | | | | | |
|---|---|---|---|---|---|---|---|---|---|---|---|---|---|
| | Ar→Cl | Ar→Pr | Ar→Rw | Cl→Ar | Cl→Pr | Cl→Rw | Pr→Ar | Pr→Cl | Pr→Rw | Rw→Ar | Rw→Cl | Rw→Pr | Mean |
| **Source-use** | | | | | | | | | | | | | |
| DAN [38] | 43.6 | 57.0 | 67.9 | 45.8 | 56.5 | 60.4 | 44.0 | 43.6 | 67.7 | 63.1 | 51.5 | 74.3 | 56.3 |
| DANN [57] | 45.6 | 59.3 | 70.1 | 47.0 | 58.5 | 60.9 | 46.1 | 43.7 | 68.5 | 63.2 | 51.8 | 76.8 | 57.6 |
| MCD [34] | 48.9 | 68.3 | 74.6 | 61.3 | 67.6 | 68.8 | 57.0 | 47.1 | 75.1 | 69.1 | 52.2 | 79.6 | 64.1 |
| CDAN+E [1] | 50.7 | 70.6 | 76.0 | 57.6 | 70.0 | 70.0 | 57.4 | 50.9 | 77.3 | 70.9 | 56.7 | 81.6 | 65.8 |
| SAFN [59] | 52.0 | 71.7 | 76.3 | 64.2 | 69.9 | 71.9 | 63.7 | 51.4 | 77.1 | 70.9 | 57.1 | 81.5 | 67.3 |
| BNM [58] | 52.3 | 73.9 | 80.0 | 63.3 | 72.9 | 74.9 | 61.7 | 49.5 | 79.7 | 70.5 | 53.6 | 82.2 | 67.9 |
| MDD [60] | 54.9 | 73.7 | 77.8 | 60.0 | 71.4 | 71.8 | 61.2 | 53.6 | 78.1 | 72.5 | **60.2** | 82.3 | 68.1 |
| BDG [61] | 51.5 | 73.4 | 78.7 | 65.3 | 71.5 | 73.7 | 65.1 | 49.7 | 81.1 | 74.6 | 55.1 | **84.8** | 68.7 |
| **Source-free** | | | | | | | | | | | | | |
| Source-only | 44.6 | 67.3 | 74.8 | 52.7 | 62.7 | 64.8 | 53.0 | 40.6 | 73.2 | 65.3 | 45.4 | 78.0 | 60.2 |
| SHOT [7] | 57.1 | **78.1** | 81.5 | 68.0 | 78.2 | 78.1 | 67.4 | 54.9 | 82.2 | 73.3 | 58.8 | 84.3 | 71.8 |
| **This paper** | **57.9** | 77.6 | **82.5** | **68.6** | **79.4** | **80.6** | **68.4** | **55.6** | **83.1** | **75.2** | 59.6 | 84.7 | **72.8** |

Table 4: Generalized SFDA accuracy (%) on Office-Home, where all the methods use a ResNet-50 backbone. Our variational model perturbation effectively adapts to the target domain while still maintaining the discriminative ability on the source domain. All per-domain results are provided in the supplementary material.

| | Office-Home | | |
|---|---|---|---|
| | Source | Target | Harmonic |
| Source-only | **83.6** | 59.1 | 69.2 |
| SHOT [7] | 71.7 | 70.4 | 71.0 |
| **This paper** | 76.6 | **71.6** | **74.0** |

## 4.4 Results

In this section, we provide the experimental results on five datasets under three different evaluation settings. We also provide more detailed results in the supplemental material.

**Offline SFDA.** We report the comparison results on Office in Table 2 and Office-Home in Table 3. From the results we observe that the variational model perturbation achieves the best average results on Office, showing the stronger generalization performance of the perturbed model. Moreover, the variational model perturbation achieves the best results on 4 out of 6 tasks. In particular, our method has improved by 1.4% compared with unperturbed SHOT. A similar trend can be observed in the Office-Home dataset, the variational model perturbation wins 9 taskes across all the 12 tasks. In addition, our method outperforms SHOT by 1.0%, which proves that the variational model perturbation can consistently and clearly improve existing methods.

**Generalized SFDA.** In Table 4 we report the results of model perturbation on Generalized SFDA tasks. In this setting, we first train the source model with 80% of the source data, and then adapt the model to the target domain. Finally, we evaluate the model on both the target domain and the remaining 20% of the source data. We observe that the performance of SHOT on the source domain decreases by 11.9% compared to the source model, indicating that SHOT forgets the knowledge learned in the source domain. Compared to SHOT, model perturbation maintains the performance on the target domain while considerably improving the performance on the source domain, revealing that our model perturbation can maintain a good balance between adaptation and knowledge preservation.

**Continual Online SFDA.** In Fig. 4 we report the test errors of different methods under the continual online SFDA, where all the methods continuously encounter 15 different types of corruption. This experiment not only tests the real-time prediction performance of the compared methods, but also shows the degree of knowledge forgetting. Note that Tent [18] only updates the parameters in the BN layers while Tent-FT (implemented by us) updates all the weights. Similar to Tent, BACS [43] also employs entropy minimization, with the difference that BACS optimizes the model through a maximum-a-posteriori probability.

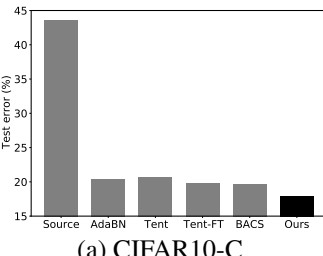 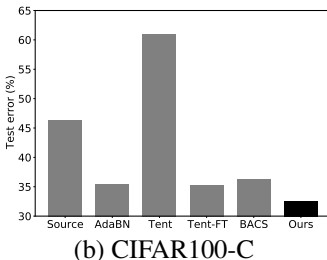 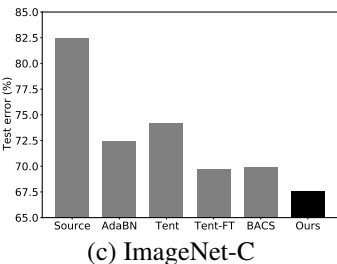

|   (a) CIFAR10-C   |   (b) CIFAR100-C   |   (c) ImageNet-C   |

Figure 4: Test errors (%) under the continual online setting. The backbone of CIFAR10-C, CIFAR100-C and ImageNet-C are WideResNet-28, ResNeXt-29 and ResNet-50. The corruption severity is 5. Model perturbations achieves the best performance across all tasks.

In Fig. 4 (a) both AdaBN and Tent have test errors larger than 20.0%, revealing that the inflexibility of freezing the weights is the bottleneck for adapting to the continuously changing target domains. By unfreezing the parameter update of the convolutional and FC layers, Tent-FT improves Tent by 0.9%. Compared to Tent and Tent-FT, the implementation of model perturbation based on Tent achieves performance advantages of 2.8% and 1.9%, respectively, demonstrating that model perturbation is the better choice for SFDA tasks. The performance advantages stem from two aspects. First, model perturbation fixes all the weights of the convolutional and FC layers to avoid large distortion of the weights and forgetting knowledge on the source domain. Second, model perturbation introduces uncertainty into the model by slightly disturbing the weights, allowing it to adapt to continuous distribution shifts. Similar conclusions can be drawn on CIFAR100-C and ImageNet-C.

## 5  Conclusion

In this paper, we address source-free domain adaptation in a probabilistic framework by variational model perturbation. Instead of updating the network parameters by further optimization, we introduce perturbations to the weights of a pre-trained model from source domains. We learn these perturbations through variational Bayesian inference. Importantly, we prove the connection between variational model perturbation and Bayesian learning of a neural networks, which provides a theoretical guarantee for the generalization performance of model perturbation. Moreover, with the parameter sharing strategy, we substantially reduce the learnable parameters compared to a fully Bayesian neural network, making the model perturbation a flexible and lightweight framework. Our variational model perturbation offers a new, probabilistic framework for source-free domain adaptation, which achieves efficient adaptation while largely preserving the discriminative ability of the source model. The experimental results on five benchmark datasets under three different evaluation settings reveal that model perturbation offers an effective way for source-free domain adaptation and boosts the performance considerably.

## Acknowledgments and Disclosure of Funding

This work is financially supported by China Scholarship Council (CSC).

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
