# OpenReview forum: "Variational Model Perturbation for Source-Free Domain Adaptation"
_NeurIPS.cc/2022/Conference — NeurIPS 2022 Accept_

### Official Review · Reviewer_N4jQ · 2022-07-08

**Rating:** 6
**Confidence:** 4
**Soundness:** 3 good
**Presentation:** 3 good
**Contribution:** 2 fair

**Summary:**

The paper proposes a variational Bayesian inference framework to perturb source model parameters to address source-free domain adaptation.

The paper proposes framework simultaneously adapts the source model to the target domain and preserves the discriminability of the model and

The paper introduces a parameter sharing strategy to achieve more efficient adaptation.

The paper provides a theoretical connetion between Bayesain learning and the proposed method.

The paper conducts experiments on three source-free domain adaptation frameworks to verify the performance of the proposed method.

**Questions:**

As discussed in the weakness, the two limitations of fine-tuning may not be severe for the domain adaptation setting, could the authors discuss some main limitations of fine-tuning such as overfitting on the target data or limited to settings with target labeled data and cannot be applied to unsupervised domain adaptation?

Does the method only limited to the computer vision setting. In the paper, the authors only gives an instatiation of the cnovolutional layer.

**Limitations:**

The authors do not discuss the limitations of the method. Could the authors discuss this in the response or point out at which place in the paper the authors discuss this?

**Strengths And Weaknesses:**

Strength:

The general idea is interesting. Prior works on fine-tuning also take techniques to reduce the divergence between the fine-tuned model and the original. This paper directly models the perturbation, which naturally limited the divergence between the tuned model and the original model, which can simultaneously avoids catestraphic forgetting and the enables adaptation.

The authors adaptively adjust the prior distribution for different convolution kernels, which avoids the influence of the variance in convolutional kernel parameters in the pretrained model.

The connection to Bayesian learning provides a theoretical basis for the proposed method.

The paper does not introduce too many extra parameters and could be trained efficiently.

The Quantification of weight uncertainty is interesting, which shows that different layers have different uncertainty on the parameters. The observation could guide following works on source-free domain adpatation.


Weakness:

On line 3-4, source-free domain adaptation assumes that both source data and labeled target data are not available but not 'or'.

On line 38-40, the paper argus that the performance on the source domain will degrade. However, in many domain adaptation settings, only the performance in the target domain is important. For the source domain, we can directly use the model trained on the source domain to perform prediction. Thus, different from continual learning, degrading the source performance may not be a shortcoming of domain adaptation methods.

On line 40-42, the bias to a specific target domain may not be a problem for the domain adaptation to a single domain but a problem for multi-target domain adaptation. However, for multi-target domain adaptation, we can also merge all the domains and perform a single target domain adaptation.

On line 91-94, the authors defautly assume that \Delta \omega has a zero mean, which means that \Delta \omega has the highest density according to the property of the Gaussian distribution. However, \Delta \omega = 0 indicates that the source network should be directly applied to the target network, which is a clearly suboptimal solution for \Delta \omega. Why does the authors use a obviously suboptimal solution at the point of the highest density for \Delta \omega.

The parameter sharing strategy is natural and should not be considered as a significant contribution. The parameters in the same kernel has strong correlations and should not be treated separately. Thus, sharing the same perturbation is a natural choice but not a special strategy.

In section 4.3, what does parameter non-shareing mean in the paragraph 'Parameter sharing strategy'?

---

> ### Author Response · Authors · 2022-08-02
> **Response for Reviewer N4jQ Part II**
>
> **Meaning of non-sharing strategy:** By non-sharing we mean to express that we learn a different $\sigma$ for each parameter in the convolutional kernel individually, rather than the whole kernel sharing a single sigma. We have clarified this in Fig. 2 in the updated manuscript.
>
> **Limitations of fine-tuning:** The reviewer gave us two possible limitations of fine-tuning. First, overfitting to target data: this is indeed a problem. Fine-tuning will update the model parameters by backpropagation while usually we have very limited data from the target domain compared to the large amount of parameters in the source model, and therefore overfitting would happen inevitably, resulting in degraded performance in the scenario of Generalized SFDA.Second, limited settings: usually fine-tuning works well in the vanilla unsupervised domain adaptation setting where we are given labeled source data and unlabeled target data simultaneously. Previous UDA methods, e.g., CDAN [1], MCD [30] and DANN [51] use labeled source data to regularize their model to guarantee its discriminative power on the source domain. The limitation of fine-tuning is that, in SFDA, since there are no regularizations from the labeled source data (e.g. cross entropy loss, etc.), errors may accumulate and the parameters of the model may be distorted, leading to low performance in Continual Online SFDA [15].
>
> **Limited to computer vision setting:** Thank you for sharing the insight. Indeed, our model perturbation is not limited to computer vision settings. The perturbation operation is architecture-agnostic, and could be applied to various architectures though we implement convolutional and fully-connected layers in this work. We have mentioned this in Section 4 in the updated supplementary material.
>
> **Limitation of our method:** As we demonstrated in the paper, our variational model perturbation is closely related to learning a Bayesian neural network. While enjoying the better generalization ability, our method could also suffer from the limitation of high training cost. This would make it computationally expensive when the model size is very large. We have added this limitation in Section 4 in the updated supplementary material.
>
> *We hope this rebuttal further solidifies your positive outlook on the paper and we are happy to discuss with you to improve our paper. Thank you.*

---

> > ### Comment · Reviewer_N4jQ · 2022-08-08
> > **Response**
> >
> > Thanks for addressing my concerns. I have changed the score from 5->6.

---

> ### Author Response · Authors · 2022-08-02
> **Response for Reviewer N4jQ Part I**
>
> We sincerely thank you for your valuable comments and suggestions. Followings are the clarifications requested in the review. Note that the reference number is based on the updated manuscript.
>
> **Description of SFDA setting:** The reviewer is correct. In the setting of SFDA, both the source domain data and the labeled target domain data are indeed unavailable at the same time. We have corrected “or” to “and” in Abstract in the updated manuscript.
>
> **Performance on the source domain:** We agree with the reviewer that usually the performance on target domains is more important. In the more generalized SFDA scenario [11] we also care about the performance on source domain since we do not necessarily have the target domain ID available. Hence, we would not know when to deploy the source model. In this sense, we would like the model to maximally preserve the performance on the source domain. Moreover, the model will be more flexible if we are aware of the generalized SFDA setting when designing the model. We have added this discussion on line 40-46 in the updated manuscript.
>
> **Bias to a specific target domain:** We agree with the reviewer that the bias to a specific target domain may not be a problem in the traditional setting of single target domain adaptation. In the Generalized SFDA setting [11], we might encounter samples from both the source and target domain when testing. Our model perturbation better adapts to such settings. We have reworded our statement on line 40-43 in the updated manuscript to make it more specific to the Generalized SFDA setting.
>
> **Zero mean for $\Delta w$:** The reason we choose zero mean for Δw is that we want to avoid the perturbed model deviating too much from the source model so that it can adapt to the more general settings of Generalized SFDA [11] and Continual Online SFDA [15]. To do so, we  infer  the posterior over perturbations, which is defined as a zero-mean isotropic Gaussian distribution. As a result, the weights of the perturbed model will vary around the weights of the source model while being generalizable due to the uncertainty induced by non-zero variance. In essence, we expand the effective coverage of the source model to the target domain. Moreover, the choice considerably reduces the amount of learnable parameters compared to a fully learnable non zero-mean Gaussian posterior while maintaining the performance. This is consistent with the findings in [21] that by replacing 95% of the variable weights in BNN with a constant zero the model still maintains good performance. We have added this discussion on line 104-108 in the updated manuscript.
>
> **Contribution of parameter sharing strategy:** We agree with the reviewer that the parameter sharing strategy is a natural choice. Indeed the strategy is well coupled with our variational perturbation as it considerably reduces the amount of learnable parameters, which enables effective and efficient optimization. While we see it as an important technical contribution in our variational framework, we have taken the reviewer’s advice and reworded this part a bit to avoid overstatement.

---

### Official Review · Reviewer_Uyrb · 2022-07-10

**Rating:** 8
**Confidence:** 5
**Soundness:** 3 good
**Presentation:** 3 good
**Contribution:** 4 excellent

**Summary:**

In this paper, the authors propose to solve the source-free domain adaptation problem by perturbing the source model to adapt it to the target domain. The perturbations for the source model are learned through variational Bayesian inference. The authors provide a parameter sharing strategy to significantly reduce the number of learnable perturbations. Therefore, the proposed method is a flexible and lightweight framework and can be integrated with existing methods. Extensive experiments on 5 datasets under 3 different SFDA settings demonstrate the superior performance of variational model perturbations.

**Questions:**

1. My major concern is why variational model perturbation is effective? Even though the learnable parameters are very few? Clear and detailed explanations are needed.
2. In Fig. 1, using the classification accuracy as an indicator of domain shifts may be not appropriate. According to Ben-David's domain adaptation theory [7][8], it is better to use the A-distance as a proxy of the domain shifts.
3. Does model perturbation depend on a specific architecture? In addition to CNNs, the effectiveness of model perturbation on other architectures, e.g., ViT, is of interest.
4. The proposed method works under the assumption that the source data are not provided. Though I acknowledge the importance of this setting, in real-world applications it is possible to make use of some of the source data. In my opinion, model perturbation can be applied to the setting of standard source-needed domain adaptation, so could you provide results of other implementations based on the standard domain adaptation setting? (e.g., CDAN).
5. Is it safe to adopt assumption of same prior p(wt) = p(Δw) in Eq. 7?
6. I don't see the results of other implementations of model perturbations. Although these results appear in the supplementary material, I still suggest that these results are included in the main text.
7. How does the number of sample points (size of the datasets) affect performance? How many target data points are sufficient for adaptation? Does the performance advantage relative to the baseline methods vary with the size of the datasets?


References (including the reference from above section):
[1] Liang J, Hu D, Feng J. Do we really need to access the source data? source hypothesis transfer for unsupervised domain adaptation[C]//International Conference on Machine Learning. PMLR, 2020: 6028-6039.
[2] Yang S, van de Weijer J, Herranz L, et al. Exploiting the intrinsic neighborhood structure for source-free domain adaptation[J]. Advances in Neural Information Processing Systems, 2021, 34: 29393-29405.
[3] Huang J, Guan D, Xiao A, et al. Model adaptation: Historical contrastive learning for unsupervised domain adaptation without source data[J]. Advances in Neural Information Processing Systems, 2021, 34: 3635-3649.
[4] Wang D, Shelhamer E, Liu S, et al. Tent: Fully Test-Time Adaptation by Entropy Minimization[C]//International Conference on Learning Representations. 2020.
[5] Zhou A, Levine S. Bayesian Adaptation for Covariate Shift[J]. Advances in Neural Information Processing Systems, 2021, 34: 914-927.
[6] Wang Q, Fink O, Van Gool L, et al. Continual test-time domain adaptation[C]//Proceedings of the IEEE/CVF Conference on Computer Vision and Pattern Recognition. 2022: 7201-7211.
[7] Ben-David S, Blitzer J, Crammer K, et al. Analysis of representations for domain adaptation[J]. Advances in neural information processing systems, 2006, 19.
[8] Ben-David S, Blitzer J, Crammer K, et al. A theory of learning from different domains[J]. Machine learning, 2010, 79(1): 151-175.

**Limitations:**

1. Line 170 and 199: typos "adaption".
2. The upper case and lower case are not consistent: "KL-divergence" in line 95 and "KL-Divergence" in line 110, "Adaptive prior" in line 126 and "adaptive prior" in line 268.
3. In Eq. 8, "w" appears for the first time. It should be "wt" according to Eq. 2.

**Strengths And Weaknesses:**

1. A flexible and lightweight framework, easy to be integrated with previous methods.
2. Very few learnable parameters.
3. It decouples the process of domain adaptation into the invariant part (the static source model) and varying part (the learnble perturbations).
4. The authors connected variational model perturbation with Bayesian neural networks, which provides a theoretical guarantee for the generalization performance of model perturbation.
5. Most experimental details required for reproducing the results were given. Code was submitted as well, which is noteworthy.
6. The performance is competitive. Variational model perturbation can achieve the best results in almost settings against the compared baselines.
7. The experiments take into account both SFDA and test time adaptation settings, which seem very sufficient. Note that previous methods solved SFDA and test time adaptation independently, e.g., SHOT[1] (SFDA), NRC[2] (SFDA), HCL[3] (SFDA), TENT[4] (test-time adaptation), BACS[5] (test-time adaptation), CoTTA [6] (test-time adaptation). This paper essentially unifies the two settings into one framework.

---

> ### Author Response · Authors · 2022-08-02
> **Response for Reviewer Uyrb**
>
> Thank you for your encouraging review. The clarifications requested in the review are as follows. Note that the reference number is based on the updated manuscript.
>
> **Explanations for effectiveness of model perturbation and parameter sharing:** Our variational model perturbation is effective mainly because of the probabilistic modeling. In this way, we perturb a deterministic model into a Bayesian model. As a result, the perturbed model can be generalized to more domains while maintaining the performance on the source domain, i.e., we expand the effective coverage of the source model. It is effective even though learnable parameters are very few because of the parameter sharing strategy. By leveraging the correlation of weights, the model parameters are largely reduced while still maintaining learning ability [21]. We have added the discussion in Section 4 in the updated supplementary material.
>
> **A-distance as indicator of domain shifts:** We have replaced Acc with A-distance in Fig. 1 in the updated manuscript. Thank you for your suggestion.
>
> **Dependence on specific architectures:** The idea of variational perturbation does not depend on specific architectures though we implement it with convolutional and fully-connected layers in this work. It is worth noting that the parameter sharing strategy in our method is specific for  the convolutional kernel. To enable parameter sharing for other non-convolutional architectures, we will need to redesign different  strategies. To directly benchmark with existing SFDA methods, which are mainly based on CNNs for computer vision tasks, we implement our idea with CNNs. We would like to take the reviewer’s advice and implement the variational perturbation idea with other architecture, e.g., ViT in our future work. We have added this discussion  in Section 4 in the updated supplementary material. Thank you.
>
> **Variational model perturbation for standard domain adaptation:** Thank you for this interesting suggestion. Our variational model perturbation is specifically designed for source-free domain adaptation. That is, we have a source-trained model and perturb the model parameters using a handful of target data. In source-needed domain adaptation, we could directly use source and target data to retrain the model if source data are already available. We have added this discussion on Section 4 in the updated supplementary material.
>
> **Priors:** They are prior distributions and usually set to be noninformative normal Gaussian distributions. For practical optimization, they serve as a regularizer in the KL term. Therefore, we believe it is safe to adopt the assumption of the same prior $p(w_t) = p(\Delta w)$. Even if the prior distributions are different, the conclusion that our variational perturbation is equivalent to learning a Bayesian neural network  still holds. We have added this discussion on line 121-126 in the updated manuscript. Thank you.
>
> **Move detailed experimental results to main text:** Should our paper be accepted we will use the extra page to include more experimental results from the supplementary in the main text. Thank you for your suggestion.
>
> **Impact of dataset size on performance:** In general, the performance will be better with a larger dataset size. The sufficiency of target data depends on the domain gap: a larger gap requires more target data. For fair comparisons, we use the same number of target samples for all compared methods. The results of different datasets demonstrate that our performance advantage over baseline methods is consistent under different sizes of datasets.
> To further verify the performance advantage of our method under different dataset sizes, we have carried out additional experiments. Specifically, in the target domain adaptation phase, we randomly selected 20%, 40%, 60% and 80% samples from the target domains in the OfficeHome dataset for adaptation. In the testing phase, we selected all the target samples (100%) for testing. The experimental results are shown below. Each result is the average of the results of five data partitioning schemes under different random seeds. From the experimental results, we conclude that our method consistently maintains performance advantage over the baselines under conditions of different dataset sizes.
>
> | Ratio | 20% | 40% | 60% | 80% |100% |
> | :---: |:---:|:---:|:---:|:---:|:---:|
> |SHOT[5]|67.2 |69.6 |70.4 |71.1 |71.8 |
> | Ours  |68.5 |71.0 |71.6 |72.3 |72.8 |
>
> **Typos and suggested references:** We have corrected all typos and also added the suggested references in the updated manuscript.
>
> *Thanks again for your positive review. We are glad to discuss with you to improve our paper.*

---

> > ### Comment · Reviewer_Uyrb · 2022-08-08
> > **Reply**
> >
> > Thanks for the rebuttal, your answers have addressed all of my main concerns!

---

### Official Review · Reviewer_66qw · 2022-07-11

**Rating:** 7
**Confidence:** 3
**Soundness:** 3 good
**Presentation:** 4 excellent
**Contribution:** 3 good

**Summary:**

The challenge of source-free domain adaptation (SFDA) is that the source data is inaccessible after the source model is trained. Two prevalent approaches to address the problem is fine-tuning the source model on target domain and incorporate the adapters into the source model and learn it with target data. However the fine-tuned model usually has poor generalization of the model since it tends to be biased to a specific target domain. The adapter/modulating method is unable to handle large domain shifts as all the convolutional layers and fully-connected layers are frozen.

To alleviate these problems, in this work, the authors proposed to perturb the parameters of the source model to achieve adaptation to target domain. The model perturbation approach learn domain-specific knowledge through variational Bayesian inference, allowing the model to generalize to relevant domains. This model-agnostic framework can efficiently adapt to target domain while largely preserving the discriminative ability of the source model. The method has been validated in various datasets and experiment settings.

**Questions:**

I only have some minor questions regrading the experiment details
- For a fair comparison, the authors adopted the same network architecture and even the same hyper-parameters (if I am wrong, please correct me) with baseline methods. Did you consider/try to tune the hyper-parameters to fully release the power of the proposed method. It would be interesting to show the best performance after tuning.
- Could you please elaborate more about why the perturbation method is able to preserve discriminative power of the source model? For instance, when the testing example comes from the source domain, how the model would perturb the model?
- In Table 2, there are 6 tasks in total, why the total winers are 10? is it because the tie existing in some tasks?
- In Table 3, the bold number in "source" column has special meaning?

**Limitations:**

I would recommend to add a pseudocode of the training algorithm in the main body or supplementary such that to help the readers to get a clear view of the model training workflow.

**Strengths And Weaknesses:**

Strengths:
- The work is well-motivated, the writing is easy to follow, and the framework is general and flexible
- Provided a novel probabilistic framework for SFDA
- Demonstrated the theoretical connection between their probabilistic framework and the Bayesian neural network
- The theoretical analysis of the method is clear and sound
- The empirical and comprehensive experiments validated the method in various settings

Weaknesses:
- To make it self-contain, a more detailed description of the model training or a pseudocode of the training algorithm is expected. For instance, in line 227 and 237, the authors only mentioned that the expected likelihood adopted the same losses used in reference papers. It would be better to include the loss function as supplementary.
- Since the authors adopted the local reparameterization trick in line 247, it would be better to briefly explain how to optimize the objective function in eq(3) so that help the general audience understand why and how to apply the trick. In turns, it also help the audience get a better understanding of the training workflow.

---

> ### Author Response · Authors · 2022-08-02
> **Response for Reviewer 66qw**
>
> We are glad that the reviewer found our paper well motivated, easy to follow, general and flexible. Followings are the clarifications requested in the review. Note that the reference number is based on the updated manuscript.
>
> **Detailed description of the model training or a pseudocode:** The detailed description of the model training is as follows:
>
> >**Input:** Unlabeled target dataset $D_t$={${x_t}|x_t \in X_t$}, pre-trained source model $G_s$ parameterized by $w_s$, the max iteration number T.
> >**Initialization:**
> > (1) Initialize the target model $G_t$ with $G_s$.
> (2) Calculate the variance Var($w_s^{li}$) in Eq. 9 for each convolution kernel of the source model layer by layer to determine the prior distribution of the convolution kernel.
> (3) Initialize the perturbation parameters $\rho$ and add $\rho$ to the optimizer.
> (4) Modify the forward function of every layer so that we can compute the perturbed weights and the corresponding output layer by layer according to Eq. 8.
> > **On training:**
> **While t < T:**
> (1) Sample a batch of target sample $x_t$ and feed them to the network $G_t$.
> (2) Compute the perturbed weights $w_t$ and the corresponding output $z_t$ layer by layer according to Eq. 8.
> (3) Compute the loss in Eq. 3 using the final output of $G_t$.
> (4) Update the perturbation parameter $\rho$ through backpropagation.
> **Output:** the perturbed model $G_t$ parameterized by $w_t$.
> **Prediction:** Classify $x_t$ by ${\tilde y}_{t}=G_t(x_t)$
>
> We have included in the supplementary material: (1) pseudo-code for the model training process; (2) loss functions of relevant methods such as SHOT [5] and Tent [16]. Thank you.
>
> **Local reparameterization trick:** We have added the detailed optimization process with local reparameterization trick in Section 3 in the supplementary material: The local reparameterization trick allows us to sample pre-activations instead of weights. Specifically, for the perturbed weight $w_t$ sampled from a factorized Gaussian $\mathcal{N}(w_s, \sigma^2I)$, the pre-activations $a = w_t h$ are distributed according to $\mathcal{N}(w_s h, \sigma^2 h^2I)$, where $\sigma$ is the learnable perturbation parameters, $w_s$ is the weight before perturbations, h is the output from the preceding layer. By doing so, we can reduce the sampling variance.
>
> **Network architectures and hyper-parameters:** For a fair comparison, we adopted exactly the same network architecture as the original methods (SHOT [5], Tent [16]). However, since the parameters learned by our method are perturbations instead of the weights, the hyper-parameters used in the original paper may be sub-optimal. Therefore, we need to tune the hyper-parameters from scratch. The results reported in the manuscript are based on the tuned hyper-parameters. Thank you.
>
> **Reason of preserving discriminative power of the source model:** In our model perturbation framework, the weights $w_s$ of the source model are fixed. We learn a perturbation parameter $\sigma$ so that the perturbed weight $w_t$ varies around the original weight $w_s$. By doing so, the perturbed model will not deviate much from the source domain, which can largely preserve the discriminative power of the source model while being able to generalize to the target domain.
>
> **Six tasks but ten winners in Table 2:** This is because in the task W$\to$D, there are 5 methods that achieve the same best accuracy. Please refer to the results of Table 1 in the supplementary material for the complete numbers. We have explained this better in the updated manuscript.
>
> **Special meaning of bold number in "source" column:** In Table 3, the bold number in "source" column means that this number is the result of our method. Since this result is not the best result in the "source" column, we have unbolded this number in the updated manuscript.
>
> **Suggestion of adding pseudo-code:** Thanks for your suggestion. We have added pseudo-code for the model training process in the updated supplementary material.
>
> *Thank you again for your constructive review. We are happy to discuss with you to improve our paper.*

---

> > ### Comment · Reviewer_66qw · 2022-08-10
> > **Thank you for the response**
> >
> > I appreciate the efforts made by the authors on the additional works and clarifications. Most of my concerns have been addressed. Thank you.

---

### Official Review · Reviewer_ZFKC · 2022-07-11

**Rating:** 7
**Confidence:** 3
**Soundness:** 3 good
**Presentation:** 3 good
**Contribution:** 3 good

**Summary:**

This paper proposes a novel model perturbation-based method for the source-free domain adaptation (SFDA) problem. Under the mild assumption, the authors connected their variational model perturbation with the bayesian neural network and proposed a perturbation parameters update method for source-free adaptation, avoiding changing the source domain model’s weight severely while improving the model’s generalizability. The experimental results on several domain adaptation benchmarks under three different settings(offline SFDA, Generalized SFDA, and continual SFDA) verify the effectiveness of the proposed method.

**Questions:**

Please refer to the questions in Weaknesses.

**Limitations:**

The authors discussed the limitations in section 2.

**Strengths And Weaknesses:**

Strengths

1. As far as I am aware, this is the first time that the probabilistic model approach is introduced in the source-free domain adaptation. The proposed method is technically sound and novel in the SFDA problem. The intuition behind the variational model perturbation method in this paper fits well the SFDA setting, which can bring some novel insights to the community.
2. The proposed method is efficient and can be combined with other SFDA methods.
3. The experiments cover three different SFDA scenarios, and the results generally look promising. The analysis in section 4.3 is comprehensive and addresses some of my concerns during reviewing.

Weaknesses

1. The Methodology section lacks some necessary description of the proposed method and generated algorithm, which results in unclarity and ambiguity. For example:
    1. What is the form of the specific objective function? （Under the assumption of prior and posterior distribution mentioned in the paper）Could the author further clarify the expression of the regularization term?
    2. Is the main parameter \sigma calculated and updated (if any) directly by the neural network？
2. Some descriptions of the experimental section are confusing.
    1. For example, in lines 218-225, while describing the offline SFDA, the paper utilizes the “training phase” represents the target training process; however, it is written “the model is first trained using 80% source data and then tested using the rest of the source data and all the target data” for Generalized SFDA setting. Then, in lines 245-247, it mentions only the training phase and the test phase. I suppose these two phrases are both noted for target training. Could the authors harmonize “source training” or “target training” to avoid ambiguity? Besides, could the source model generation process be introduced to make the experimental description more complete? Considering the particular setting of SFDA, I look forward to a clearer elaboration of the experimental details.
3. The experimental results only mention the accuracy but do not include the variance part. As the proposed model perturbation method only optimized the perturbation’s variance and did not update the source model’s weight, I am curious about the stability of the proposed method.
4. There are some unclear mathematical expressions. For example, in line 110-111, “Here we assume that we adopt the same prior, i.e., $p(w_t)=p(\Delta w)$”, does it means that these two random variables have the same variance?
5. In line 91, this paper directly assumes the zero-mean isotropic Gaussian distribution as the posterior distribution family. Could the authors explain more about such choices and assumptions?

================================================ ================================================

Thanks for the effort from the authors, and the rebuttal has well addressed my concerns mentioned above. I would like to raise my score to 7.

---

> ### Author Response · Authors · 2022-08-02
> **Response for Reviewer ZFKC Part II**
>
> **Descriptions of the experimental section:** In Section 6 of the updated supplementary material, we have clarified and standardized the relevant descriptions throughout the manuscript. For a clearer elaboration of the experimental details, we clarify SFDA consists of three phases, namely, *source pre-training*, *target adaptation* and *testing*.
> Our paper considers three different settings, each with a different experimental procedure.
> *Offline SFDA [3,4]:* in the source pre-training phase, we use all labeled source domain data to train the source model by minimizing the cross entropy loss. In the target adaptation phase, we use all unlabeled target data to adapt the model to the target domain. In the testing phase, we make predictions for all target data.
> *Generalized SFDA [11]:* we first split the source domain data into 80% and 20% parts. In the source pre-training phase, the data of the 80% part are labeled and we use them to pre-train the source model. In the target adaptation phase, we use all the unlabeled target data to adapt the model to the target domain. In the testing phase, we predict the remaining 20% source data and all target data.
> *Continual Online SFDA [16]:* the source domain consists of the clean images from Cifar10, Cifar100 and ImageNet, while the target domain consists of the corrupted images from Cifar10-C, Cifar100-C and ImageNet-C. In the source pre-training phase, we directly adopt the publicly available model pre-trained on the clean training set as the source model for SFDA. Specifically, Cifar10-C uses WideResNet-28 from RobustBench. Cifar100-C uses ResNeXt-29 from [48]. ImageNet-C uses ResNet-50 from RobustBench. Different from the previous settings, the target adaptation phase and the testing phase in this setting are carried out simultaneously. Specifically, the corrupted images are fed into the network in an online fashion. At each iteration, we first predict the corrupted images and then update the model for one step. We minimize the entropy loss just as Tent [16]. Since there are 15 different corruption types, we adapt the source pre-trained model to each corruption type sequentially.
>
> **The variance part of the experimental results:** We have added the variance part of the experimental results in the updated supplementary material. From the results, we observe that our method maintains consistent performance on all tested tasks.
>
> **Priors:** The priors serve as a regularizer in practical optimization while our goal is to infer the posteriors. Therefore the two inferred variables in the posteriors do not necessarily have the same variance. We have updated this discussion on line 121-126 in the updated manuscript.
>
> **Choice of zero-mean isotropic Gaussian distribution:** We would like the weights of the perturbed model to vary around the weights of the source model while being more generalizable due to the uncertainty induced by non-zero variance. In doing so, the model can, to a large extent, preserve the discriminative ability on the source domain. Then, model perturbation can perform well in more general settings, i.e. Generalized SFDA [11] and Continual Online SFDA [15]. Moreover, the choice considerably reduces the amount of learnable parameters compared to a fully learnable non zero-mean Gaussian posterior while maintaining the performance. This is consistent with the findings in [21] that by replacing 95% of the variable weights in a BNN with a constant zero the model still maintains good performance. We have updated this discussion on line 104-108 in the updated manuscript.
>
> *Thanks again for your positive review. We are happy to discuss with you to improve our paper.*

---

> > ### Comment · Reviewer_ZFKC · 2022-08-08
> > **Thanks for the rebuttal**
> >
> > Thank you for responding to my problems with this paper.
> >
> > I have read the authors' responses and the updated manuscript. The added details about the algorithm and the experimental part have well addressed my concerns. With a clear presentation and rich experiments, I think this paper could provide novel ideas for the source-free unsupervised domain adaptation problems, and I will update my score accordingly.

---

> ### Author Response · Authors · 2022-08-02
> **Response for Reviewer ZFKC Part I**
>
> Thank you for your constructive review and suggestions. The clarifications requested in the review are as follows. Note that the reference number is based on the updated manuscript.
>
> **Specific objective function:** On line 96-103 of the updated manuscript, we have clarified that in variational model perturbation the learnable parameters are the standard deviations of the perturbations $\sigma$. Hence, the specific objective function we minimize is:
> $$\mathcal{L} = KL[q(\Delta w|0,\sigma)||p(\Delta w)] - E_{q(\Delta w|0,\sigma)} [p(D_t|\Delta w)]$$
> which includes a KL divergence loss and an expectation likelihood loss.
> Note that the specific form of the second term depends on the model being perturbed. For SHOT[5], maximizing the expectation term is implemented as minimizing the information loss and self-supervised loss. For Tent[16], it is implemented as minimizing the entropy loss.
>
> **Clarification of the expression of the regularization term:** The KL divergence is defined as:
> $$KL[q(x)||p(x))=\int q(x) log(\frac{q(x)}{p(x)}] dx,$$
> where q(x) and p(x) are probability densities of the variational posterior and prior distributions, respectively. Assume $q(\Delta w|0,\sigma)\sim \mathcal{N}(\mu_0, \sigma_0^2 \,\mbox{I})$, $p(\Delta w)\sim\mathcal{N}(\mu_1, \sigma_1^2 \, \mbox{I})$, then the KL divergence loss can be calculated as:
> $$\mathrm{KL}[q(\Delta w|0,\sigma)||p(\Delta w)] = \frac{1}{2}[\mbox{tr}(\sigma_1^{-1} \sigma_0) - k + (\mu_1-\mu_0)^{\rm T} \sigma_1^{-1}(\mu_1-\mu_0) + \mbox{log}(\frac{\mbox{det} \sigma_1}{\mbox{det} \sigma_0})],$$ where $k$ is the dimension of the random variable, tr and det represent trace and determinant operation, respectively.
>
> **Whether $\sigma$ is calculated and updated directly:** On line 127-133 of the updated manuscript, we have mentioned that to ensure $\sigma$ is non-negative, in practical implementations $\sigma$ is calculated as $\sigma=\sqrt{exp(\rho)}$, where $\rho$ is the *de facto* variational posterior parameter. We calculate and update $\rho$ by the neural network.
> To avoid unclarity and ambiguity, we have modified these descriptions in the updated manuscript and supplementary material. Thank you.

---

### Author Response · Authors · 2022-08-08
**Shared response to all reviewers**

We appreciate your positive reviews and support. We are glad that the rebuttal and additional experiments adequately addressed your concerns. Finally, thanks for your acknowledgment of the value of our work, including the ideas, experimental results, and the potential impact both in research and real-world applications.

---

### Meta-Review · Area_Chair_dbLo · 2022-08-26

**Recommendation:** Accept
**Confidence:** Certain

**Metareview:**

This paper proposes a novel probabilistic framework for source-free domain adaptation, in which the source model serves as the invariant part (mean) while a perturbation (variance) is applied to the source model parameters to derive the target model that accounts for the target-specific distribution. All four reviewers provided detailed and constructive comments, which were well taken into account by the authors in their revision and rebuttal. After discussion, all reviewers were positive about the paper. AC agreed with the reviewers that this paper introduces a novel, solid, and parameter-simple approach to source-free domain adaptation with comprehensive empirical evaluation for several settings, which will be widely interested by the community. A further comment of AC is that the connection to Shai Ben-David et al.'s seminal bound is rather off-topic to this paper, making the discussion subject to flaw --- there is no formal modeling of the source and target data distributions that is required by the bound, while the bound cannot describe domain relatedness in terms of model parameters. So I suggest the authors to remove this part to make the paper more convincing.

**Award:**

No

---

### Decision · Program_Chairs · 2022-09-14

Accept